# Image Understanding Makes for A Good Tokenizer for Image Generation

**Luting Wang**[*†]  **Yang Zhao**[1†]  **Zijian Zhang**[1]  **Jiashi Feng**[1]  **Si Liu**[‡]  **Bingyi Kang**[1‡§]

[1]ByteDance

## Abstract

Modern image generation (IG) models have been shown to capture rich semantics valuable for image understanding (IU) tasks. However, the potential of IU models to improve IG performance remains uncharted. We address this issue using a token-based IG framework, which relies on effective tokenizers to map images into token sequences. Currently, *pixel reconstruction* (*e.g.*, VQGAN) dominates the training objective for tokenizers. In contrast, our approach adopts the *feature reconstruction* objective, where tokenizers are trained by distilling knowledge from pretrained IU encoders. Comprehensive comparisons indicate that tokenizers with strong IU capabilities achieve superior IG performance across a variety of metrics, datasets, tasks, and proposal networks. Notably, VQ-KD$_{\text{CLIP}}$ achieves 4.10 FID on ImageNet-1k. Visualization suggests that the superiority of VQ-KD can be partly attributed to the rich semantics within the VQ-KD codebook. We further introduce a straightforward pipeline to directly transform IU encoders into tokenizers, demonstrating exceptional effectiveness for IG tasks. These discoveries may energize further exploration into image tokenizer research and inspire the community to reassess the relationship between IU and IG. The code is released at `https://github.com/magic-research/vector_quantization`.

## 1 Introduction

Image understanding (IU) and image generation (IG) have been the core pursuits of computer vision research for a long time. Thanks to the progress in generative models [15,35–37,48] and network architectures [9,43], IG has witnessed remarkable advancements in recent years. These advancements spurred extensive research on leveraging powerful IG models for IU tasks (Fig. 1). Studies have shown that IG models can benefit IU tasks in various ways, including data augmentation through synthetic data generation [45, 46, 51], improved representation learning [16, 28, 52], and utilizing intermediate features from IG models for solving perception tasks [26, 53]. However, the reciprocal question remains largely uncharted: *how might IU models aid IG tasks?*

The primary focus of this paper lies in the AutoRegressive (AR) IG framework, which is gaining considerable attention for its excellence in generating high-quality images and videos [20, 48, 50]. This framework operates in a two-stage process. The first stage learns a tokenizer to map images into sequences of discrete tokens. Subsequently, the second stage trains a proposal network to model the token sequences. As underlined by prior research [4, 49], the quality of the tokenizers significantly influences overall IG performance. Meanwhile, tokenizers and IU encoders adhere to a similar structure as they both aim to map images into latent representations, either discrete or continuous. As a result, the token-based IG framework provides an optimal environment for investigating the relationship between IU and IG. Through comprehensive studies, we demonstrate

---

[*]Work done during an internship at ByteDance. Email: wangluting@buaa.edu.cn.

[‡]Corresponding authors. Email: liusi@buaa.edu.cn, bingyikang@bytedance.com.

[†]Equal contribution.     [§]Project lead.

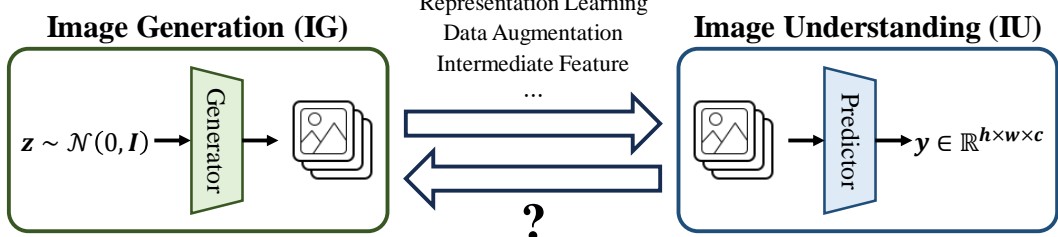

Figure 1: Extensive studies have tried to adopt IG models for IU. However, few attempts have been made to use IU models in IG.

that existing IU models from representation learning can be useful in generative models, even if they are not specifically designed for the IG task.

Our study involves training three components within the AR framework: tokenizer, decoder, and proposal network. Traditionally, *pixel reconstruction* has been the dominant objective for training tokenizers, such as VQGAN [10] and FSQ [25]. To the best of our knowledge, we are the first to systematically demonstrate that *feature reconstruction* (VQ-KD [27]) achieves better IG performance. [1] This approach distills knowledge from pretrained IU encoders to tokenizers. Therefore, the training strategy of the IU encoder is crucial for the performance of the tokenizer. In this regard, we investigate four representative IU encoders: ViT [9], CLIP [29], DINO [3], and MAE [13]. Following VQGAN [10], we train decoders to restore pixel values from discrete tokens, and proposal networks (AR or NAR) that can model the distribution of image tokens. The models are then evaluated using various metrics, including codebook usage, Fréchet Inception Distance (FID) [14], Inception Score (IS) [34], perplexity (PPL), *etc*.

Initially, we compare the above tokenizers on IN-1k for class-conditional IG. VQ-KD achieves $4.10$ $\text{FID}_{\text{AR}}$, outperforming VQGAN ($15.78$ $\text{FID}_{\text{AR}}$) by a large margin. FSQ experiments confirm that the superiority of VQ-KD is not solely attributable to the specific quantization operation or high codebook usage. More generally, VQ-KD consistently outperforms across different proposal networks, datasets, and tasks.

We analyze VQ-KD from multiple perspectives. By visualizing the codebook, we discover that codes from VQ-KD carry more semantics than VQGAN, which makes them easier to model and subsequently improve the IG quality. Building upon this insight, we propose a straightforward pipeline to efficiently transform IU encoders into tokenizers, outperforming VQ-KD on the MS-COCO dataset. We also find that tokenizers with weaker IU capabilities require larger proposal networks for effective AR modeling and show less robustness to variations in the training images. Finally, we conduct qualitative analysis to present the visual results.

In sum, the key insights from our study include the following: 1) This research is the first to demonstrate that IU models can substantially enhance IG through VQ-KD; 2) Tokenizers with strong IU capabilities consistently outperform conventional VQGAN-based methods across various metrics, datasets, tasks, and network architectures; 3) The VQ-KD codebook encapsulates more semantics than VQGAN, contributing to the superiority of VQ-KD in IG.

We believe these findings can benefit future research on image tokenizers and provoke further discussion on the relationship between IU and IG.

## 2 Related Work

**Image Tokenization.** Vector Quantization (VQ) [11] is originally developed for data compression. To circumvent posterior collapse in the VAE [19] framework, VQ-VAE [41] adopts VQ for image tokenization. Subsequently, VQGAN [10] introduces adversarial and perceptual losses to enhance the quality of the generated images.

Vanilla VQGAN suffers from limitations like low codebook usage, limited semantic representation ability, and the trade-off between modeling efficiency and image quality. To address these

---

[1]VQ-KD was initially proposed for image pretraining.

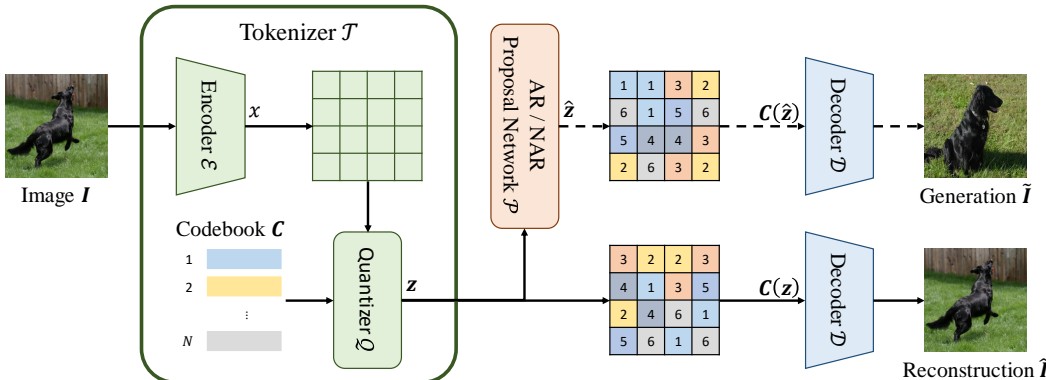

Figure 2: The token-based IG framework. Solid and dashed lines represent training and inference pipelines, respectively. During training, the tokenizer $\mathcal{T}$ tokenizes an image $\mathbf{I}$ into discrete codes $\mathbf{z}$. A proposal network $\mathcal{P}$ is trained to model the distribution $p(\mathbf{z})$, while a decoder $\mathcal{D}$ learns to reconstruct $\mathbf{I}$. During inference, we sample codes $\hat{\mathbf{z}}$ from $\mathcal{P}$, which guides $\mathcal{D}$ to perform generation.

challenges, researchers have focused on improving the codebook. ViT-VQGAN [47] adopts a ViT-based [9] autoencoder to create more expressive code vectors. SeQ-GAN [12] improves the perceptual loss and decoder to balance between semantic compression and detail preservation. SQ-VAE [40] improves VQ-VAE with stochastic quantization and a trainable posterior categorical distribution. VQ-WAE [44] builds upon SQ-VAE by encouraging the discrete representation to be a uniform distribution via a Wasserstein distance. HQ-VAE [39] employs random re-initialization of inactive code vectors. CVQ-VAE [55] selects encoded features as anchors to update dead codes. VQ-KD [27] adopts knowledge distillation instead of image reconstruction as the objective to train VQ-VAE. LFQ [49] and FSQ [25] adopt bounded scalar quantization techniques from neural compression to harness the potential of extra-large codebooks.

Furthermore, several works explore the potential of multiple codebooks. VQ-VAE-2 [32] extends VQ-VAE to a multi-scale hierarchical organization. RQ-VAE [21] and MoVQ [54] aim to represent each feature as a stack of tokens, where RQ-VAE adopts an iterative way to factorize features into a series of residuals and MoVQ models features across multiple channels via specialized modulation.

**Token-based Image Generation.** Inspired by the success of GPT [2, 30, 31], VQ-VAE and most of its derivative works [10, 54] adopt AR transformers to model the token sequence. This approach leverages techniques from text generation to enhance IG performance. However, the decoding time of AR models scales linearly with the length of the token sequence. To accelerate decoding, MaskGIT [5] introduces a bidirectional transformer, referred to as the NAR proposal network.

Given the versatility of token-based modeling, both AR and NAR proposal networks can be easily extended to conditional IG scenarios. For instance, VQGAN uses a class token as the condition in its AR proposal network for class-conditional IG. With an NAR proposal network, MUSE [4] adopts text embeddings to predict masked image tokens in Text-to-Image generation.

## 3 Token-Based Image Generation

We start with the two-stage IG framework in Sec. 3.1. Subsequently, Sec. 3.2 details the architecture and training protocol for the tokenizers under consideration. Sec. 3.3 explains the evaluation benchmark. Sec. 3.4 further outlines our main observations derived from the IN-1k experiments. Lastly, we validate the observations under different settings in Sec. 3.5.

### 3.1 Two-Stage Image Generation

We illustrate the two-stage IG framework in Fig. 2. Given an image $\mathbf{I} \in \mathbb{R}^{H \times W \times 3}$, the encoder $\mathcal{E}$ converts this image into a feature map $x \in \mathbb{R}^{h \times w \times d}$, where $(h, w) = (H/f, W/f)$ and $f$ is a downsample factor. Let a codebook $\mathbf{C}$ be a set of $N$ code vectors $\{c_i\}_{i=1}^{N} \in \mathbb{R}^{N \times d}$, where each code vector $c_i \in \mathbb{R}^d$ corresponds to a specific code $i$. A quantizer $\mathcal{Q}$ then maps $x$ into a sequence

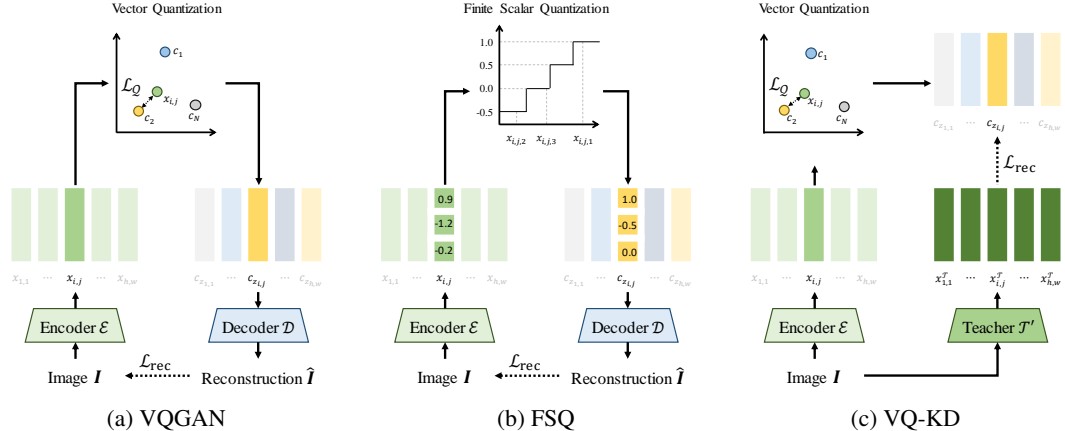

Figure 3: The architecture and training objective of different image tokenizers.

of codes $\mathbf{z} = \{z_i\}_{i=1}^{L}$, where $L = h \times w$ defines the sequence length and $z_i$ is an integer that falls within the range $[1, N]$. Let $c_{z_i}$ denote the code vector that corresponds to code $z_i$. Similarly, $\mathbf{C}(\mathbf{z}) = \{c_{z_i}\}_{i=1}^{L} \in \mathbb{R}^{L \times d}$ represents a sequence of code vectors associated with the code sequence $\mathbf{z}$. The encoder $\mathcal{E}$, quantizer $\mathcal{Q}$, and codebook $\mathbf{C}$ collectively form an image tokenizer $\mathcal{T}$.

The proposal network $\mathcal{P}$ models the distribution over $\mathbf{z}$, where the distribution is denoted as $p(\mathbf{z})$. Early proposal networks are implemented as an AR transformer, which sequentially models $p(z_i|z_{1:i-1})$ and formulates $p(\mathbf{z})$ as $\prod_{i=1}^{h \times w} p(z_i|z_{1:i-1})$. While the AR transformers can be trained in parallel, it has to sequentially decode $\mathbf{z}$ during inference, which renders it inefficient. Therefore, NAR proposal networks are being prevalent [5], which typically adopt bidirectional transformers to model $\mathbf{z}$. We denote the two types of proposal networks as $\mathcal{P}_{\text{AR}}$ and $\mathcal{P}_{\text{NAR}}$, respectively.

Finally, a decoder $\mathcal{D}$ maps the code vectors to the pixel space. In training, $\mathcal{D}$ takes $\mathbf{C}(\mathbf{z})$ as input and learns to reconstruct the original image $\mathbf{I}$ as $\hat{\mathbf{I}} \in \mathbb{R}^{H \times W \times 3}$. In inference, a sequence of codes $\hat{\mathbf{z}}$ is sampled from $p(\mathbf{z})$, translated to $\mathbf{C}(\hat{\mathbf{z}})$, and then fed into $\mathcal{D}$ to generate an image $\tilde{\mathbf{I}}$.

## 3.2 Image Tokenizers

In this paper, we focus on three types of image tokenizers: VQGAN [10], FSQ [25], and VQ-KD [27].

Let $x_{i,j} \in \mathbb{R}^d$ be a vector in the feature map $x$. As shown in Fig. 3a, to quantize $x_{i,j}$, the VQGAN tokenizer looks up the codebook $\mathbf{C}$ for the closest code vector in terms of Euclidean distance:

$$z_{i,j} = \arg\min_{z} \|x_{i,j} - c_z\|_2. \tag{1}$$

Since the quantization process is non-differentiable, VQGAN adopts the Straight-Through Estimator (STE) [1] to optimize the encoder $\mathcal{E}$, which copies gradients from $\mathbf{C}(\mathbf{z})$ to $x$. As a result, the codebook $\mathbf{C}$ receives no gradient. To optimize $\mathbf{C}$, VQGAN introduces a quantization loss $\mathcal{L}_{\mathcal{Q}}$:

$$\mathcal{L}_{\mathcal{Q}} = \|\text{sg}[x] - \mathbf{C}(\mathbf{z})\|_2^2 + \beta\|x - \text{sg}[\mathbf{C}(\mathbf{z})]\|_2^2, \tag{2}$$

where $\text{sg}[\cdot]$ denotes the stop-gradient operation and $\beta$ is the loss weight. The first term is the codebook loss, which optimizes the codebook. The second term is the commitment loss to make sure the encoder $\mathcal{E}$ commits to a code vector [41]. Therefore, the overall loss for VQGAN is defined as:

$$\mathcal{L} = \mathcal{L}_{\mathcal{Q}} + \mathcal{L}_{\text{rec}}(\mathbf{I}, \hat{\mathbf{I}}), \tag{3}$$

where $\mathcal{L}_{\text{rec}}$ is the reconstruction loss between image $\mathbf{I}$ and reconstruction $\hat{\mathbf{I}}$, which includes $\ell_1$ loss, perceptual loss, and adversarial loss.

Based on VQGAN, FSQ introduces a simpler image tokenizer, without the need for codebook lookup and quantization loss. As shown in Fig. 3b, FSQ adopts finite scalar quantization to quantize

Table 1: Comparison between image tokenizers on IN-1k.

| Tokenizer $\mathcal{T}$ | Codebook Usage (%) | rFID $\downarrow$ | $\mathcal{P}_{AR}$ | | | $\mathcal{P}_{NAR}$ | |
|---|---|---|---|---|---|---|---|
| | | | PPL $\downarrow$ | $FID_{AR} \downarrow$ | $IS_{AR}$ | $FID_{NAR} \downarrow$ | $IS_{NAR}$ |
| VQGAN | 4.9 | 5.09 | 116.75 | 24.11 | 39.52 | 20.03 | 48.30 |
| FSQ | **100.0** | 4.96 | 791.56 | 40.17 | 26.40 | 29.78 | 33.63 |
| VQ-KD$_{CLIP}$ | **100.0** | 4.96 | **53.73** | 11.78 | **128.18** | 9.51 | **121.33** |
| VQ-KD$_{ViT}$ | **100.0** | 3.69 | 89.30 | **11.40** | 107.56 | **8.45** | 108.75 |
| VQ-KD$_{DINO}$ | **100.0** | **3.41** | 74.07 | 13.15 | 80.89 | 10.21 | 91.39 |
| VQ-KD$_{MAE}$ | **100.0** | 4.93 | 280.06 | 26.85 | 40.03 | 16.11 | 59.05 |

each channel of $x_{i,j}$ into a finite set of scalars. Since the quantization process involves no trainable parameter, FSQ can be trained with solely the reconstruction loss $\mathcal{L}_{rec}(\mathbf{I}, \hat{\mathbf{I}})$.

Unlike VQGAN and FSQ, which are designed for IG, VQ-KD was originally presented in BEiT v2 to provide supervision for IU models. As shown in Fig. 3c, VQ-KD is trained to reconstruct the feature map $x^{\mathcal{T}}$ encoded by a pretrained teacher $\mathcal{T}'$. Formally, the reconstruction loss is defined as:

$$\mathcal{L}_{rec} = -\cos\left(\sigma(\mathbf{C}(\mathbf{z})), x^{\mathcal{T}}\right), \tag{4}$$

where $\cos(\cdot, \cdot)$ represents cosine similarity and $\sigma$ is a feature adapter. $\sigma$ is implemented as a decoder, which maps $\mathbf{C}(\mathbf{z})$ to the same feature space as $x^{\mathcal{T}}$.

In this study, we examine VQ-KD using four types of pretrained teachers, including fully-supervised, text-supervised, contrastive, and Masked Image Modeling (MIM). We use VQ-KD$_{CLIP}$ and VQ-KD$_{DINO}$ to represent VQ-KD tokenizers trained with CLIP [29] and DINO [3] teachers, respectively. VQ-KD$_{MAE}$ and VQ-KD$_{ViT}$ represent tokenizers trained with MAE [13] and ViT [9] teachers. The latter two teachers are pretrained on IN-1k utilizing a ViT-B/16 architecture.

## 3.3 Benchmark

We detail how we fairly compare different tokenizers for token-based IG here.

For each tokenizer, we train a proposal network $\mathcal{P}$ and a decoder $\mathcal{D}$ to constitute an image generator. In training, the tokenizer is frozen to ensure fairness. Thus, $\mathcal{P}$ and $\mathcal{D}$ can be trained in parallel. We follow VQGAN [10] to train the AR proposal network and the decoder. The NAR proposal network is trained following MAGE [22]. Implementation details can be found in Appendix B.

Our benchmark adopts various metrics to comprehensively evaluate the image tokenizers. Given an image tokenizer, we assess the effectiveness of its encoding process by evaluating the codebook usage. To assess the generative capabilities of the image tokenizers, we evaluate IS [34] and FID [14] on the generated images $\tilde{\mathbf{I}}$. We assess the reconstruction capabilities of the image tokenizers by reporting the reconstruction FID (rFID). In addition, we present the PPL scores to appraise the AR modeling proficiency of the image tokenizers. A low PPL score implies that $\mathcal{P}_{AR}$ easily models $\mathbf{z}$. Details about the evaluation metrics can be found in Appendix C.

## 3.4 Main Observation

We evaluate the *class-conditional IG* performance of VQGAN, FSQ, and VQ-KD tokenizers on IN-1k. The results in Tab. 1 leads to the following observations.

**VQ-KD significantly enhances generation quality over VQGAN.** Equipped with either AR or NAR proposal networks, VQ-KD tokenizers consistently outperform VQGAN and FSQ, as evidenced by superior FID and IS metrics. In particular, VQ-KD$_{ViT}$ attains an $FID_{AR}$ of $11.40$ and an $FID_{NAR}$ of $8.45$, both less than half of those from VQGAN ($24.11$ $FID_{AR}$ and $20.03$ $FID_{NAR}$).

Tab. 2 presents a system-level comparison between VQ-KD$_{CLIP}$ and other class-conditional IG models on IN-1k at a resolution of $256 \times 256$. With a 1.4B AR proposal network, VQ-KD$_{CLIP}$ achieves an FID of $4.10$, surpassing prior AR, NAR, and several VQ-based diffusion models.

Table 2: System level comparison on IN-1k.

| Model | Architecture | #params | FID ↓ |
|---|---|---|---|
| VQGAN [10] | AR | 1.4B | 15.78 |
| RQ-VAE [21] | AR | 1.4B | 8.71 |
| ViT-VQGAN [47] | AR | 1.7B | 5.30 |
| MoVQ [54] | NAR | 307M | 8.78 |
| MaskGIT [5] | NAR | 227M | 6.18 |
| FSQ [25] | NAR | 227M | 4.53 |
| LDM-8-G [33] | Diffusion | 506M | 7.76 |
| CVQ-VAE [8] | Diffusion | 400M | 6.87 |
| VQ-KD$_{CLIP}$ [27] (ours) | AR | 1.4B | **4.10** |

**The superiority of VQ-KD is irrelevant to the quantization operation and codebook usage.** Both VQ-KD$_{CLIP}$ and FSQ record $100.0\%$ codebook usage and 4.96 rFID, but VQ-KD$_{CLIP}$ achieves lower FID$_{AR}$ and higher IS$_{AR}$ scores. Moreover, VQ-KD proves robustness to high codebook usage, with the PPL metric of most VQ-KD tokenizers surpassing VQGAN. In contrast, FSQ lags behind VQGAN in terms of PPL, suggesting that the high codebook usage of FSQ hinders $\mathcal{P}_{AR}$ from modeling the code sequence $\mathbf{z}$. As demonstrated in Sec. 4.1, this difference is likely due to the rich semantics in the VQ-KD feature map.

**Tokenizers with stronger semantic understanding tend to deliver superior IG performance.** Considering the FID and IS metrics, we find that VQ-KD tokenizers with supervised teachers (CLIP and ViT) consistently surpass those with unsupervised teachers (DINO and MAE). While VQ-KD$_{DINO}$ achieves the lowest rFID and PPL, its 13.15 FID$_{AR}$ is worse than VQ-KD$_{CLIP}$ (11.78) and VQ-KD$_{ViT}$ (11.40). This trend can be attributed to the superior capability of supervised models in capturing semantics compared to the unsupervised ones.

## 3.5 Further Verification

**The superiority of VQ-KD holds across proposal networks.** As seen in Tab. 1, all VQ-KD tokenizers surpass VQGAN and FSQ in the FID$_{NAR}$ and IS$_{NAR}$ metrics. In particular, VQ-KD$_{ViT}$ scores the lowest FID$_{NAR}$ at 8.45 and VQ-KD$_{CLIP}$ scores the highest IS$_{NAR}$ at 121.33. In contrast, VQ-GAN only achieves an FID$_{NAR}$ of 20.03 and an IS$_{NAR}$ of 48.30. It is also worth mentioning that VQ-KD$_{CLIP}$ and VQ-KD$_{ViT}$ show slightly better performance than VQ-KD$_{DINO}$ and VQ-KD$_{MAE}$, further supporting our conclusion that superior semantic understanding in supervised models plays a significant role in enhancing the quality of IG.

Table 3: Comparison between image tokenizers on MS-COCO. T2I experiments are conducted on the MS-COCO Captions dataset.

| Tokenizer $\mathcal{T}$ | Codebook Usage (%) | rFID ↓ | $\mathcal{P}_{AR}$ | | FID$_{T2I}$ ↓ |
|---|---|---|---|---|---|
| | | | PPL ↓ | FID$_{AR}$ ↓ | |
| VQGAN | 2.4 | 16.21 | **47.89** | 38.43 | 24.11 |
| FSQ | **100.0** | 4.62 | 1040.02 | 44.64 | 23.36 |
| VQ-KD$_{CLIP}$ | 82.2 | 5.48 | 72.31 | 29.80 | **11.17** |
| VQ-KD$_{ViT}$ | **100.0** | 3.70 | 117.10 | 23.51 | 15.49 |
| VQ-KD$_{DINO}$ | **100.0** | **2.69** | 129.93 | **17.55** | 11.50 |
| VQ-KD$_{MAE}$ | **100.0** | 3.51 | 317.98 | 44.01 | 15.60 |

**The superiority of VQ-KD holds across datasets.** We conduct *unconditional IG* experiments on the MS-COCO dataset [23], which contains images of greater complexity than IN-1k. As demonstrated in Tab. 3, VQ-KD$_{DINO}$ achieves a FID$_{AR}$ metric of 17.55, significantly outperforming VQ-GAN (38.43) and FSQ (44.64). Since the ViT teacher in VQ-KD$_{ViT}$ is pretrained on IN-1k, the rFID and FID$_{AR}$ metrics of VQ-KD$_{ViT}$ are slightly inferior to VQ-KD$_{DINO}$. Note that the PPL metric of VQGAN is misleadingly favorable, due to its low codebook usage (2.4%).

**The superiority of VQ-KD holds across tasks.** *Text-to-Image* (T2I) experiments are conducted on the MS-COCO Captions dataset [6]. As shown in Tab. 3, the FID$_{T2I}$ metric of VQ-KD tokenizers range from 11.17 to 15.60, while VQGAN and FSQ only achieves 24.11 and 23.36, respectively.

# 4 Analysis

In this section, we analyze tokenizers based on feature reconstruction from various perspectives.

## 4.1 Codebook Visualization

We delve into the superiority of VQ-KD by visualizing its codebook. From IN-1k, we randomly choose four categories: golden retriever, pirate ship, valley, and sea anemone. For each image belonging to these categories, we deploy t-SNE [42] to project the feature map $x$ and the code vectors $\mathbf{C}(\mathbf{z})$ into a two-dimensional space. $x$ is colored according to the image category and $\mathbf{C}(\mathbf{z})$ is illustrated in black. As depicted in Fig. 4, the feature space of VQ-KD shows superior organization compared to VQGAN. In the VQ-KD feature space, $x$ from the same category are grouped together. This implies that each code in the VQ-KD codebook conveys clear semantics. As codes with similar semantics are likely to present concurrently in an image, it becomes easier for the proposal network to model the code sequence $\mathbf{z}$. Conversely, each code in the VQGAN codebook is shared by multiple categories, resulting in semantics ambiguity. Hence, as illustrated in Tab. 1, the PPL metric for VQGAN is higher than VQ-KD, even though its codebook usage is considerably lower.

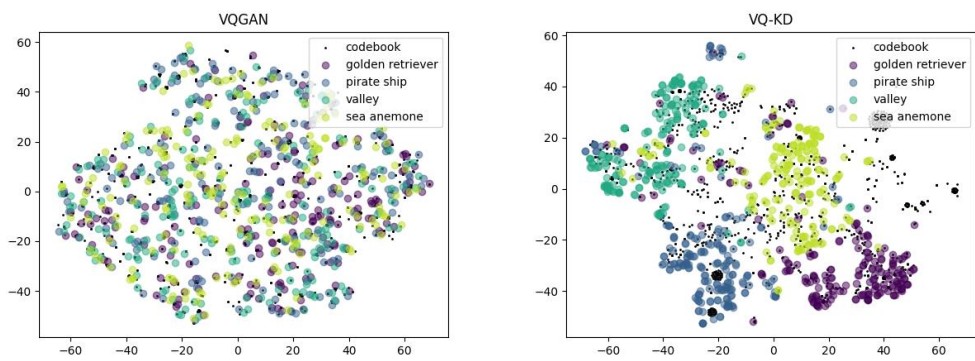

Figure 4: Codebook visualization of VQGAN and VQ-KD$_{\text{ViT}}$. Best viewed in color.

## 4.2 Clustering Pretrained Models as Tokenizers

To better harness the semantics in IU encoders, we propose a straightforward pipeline that transforms IU encoders into tokenizers via feature clustering. Given a pretrained IU model $\mathcal{T}'$, we employ it to encode the feature map $x^{\mathcal{T}}$ and subsequently utilize a clustering approach [55] to acquire $N$ clusters. The cluster centroids constitute a codebook $\mathbf{C}$. $\mathcal{T}'$ remains frozen during training, which significantly accelerates the training process. As shown in Tab. 4, Cluster$_{\text{ViT}}$ presents $13.40$ FID$_{\text{AR}}$, $10.58$ FID$_{\text{T2I}}$, and $0.245$ CLIP score on MS-COCO, outperforming all tokenizers in Tab. 3. This suggests that pretrained models with simple feature clustering can become good tokenizers. However, the cluster-based tokenizers behave worse in terms of rFID, since they encode little appearance detail in $x^{\mathcal{T}}$, which is essential for exact reconstruction. As a result, their FID and IS metrics on IN-1k are marginally weaker than those of their VQ-KD counterparts.

Table 4: Performance of cluster-based tokenizers.

| Encoder $\mathcal{E}$ | IN-1k | | | MS-COCO | | |
|---|---|---|---|---|---|---|
| | rFID ↓ | FID$_{\text{AR}}$ ↓ | FID$_{\text{NAR}}$ ↓ | rFID ↓ | FID$_{\text{AR}}$ ↓ | FID$_{\text{T2I}}$ ↓ |
| CLIP | 9.13 | 14.81 | 12.83 | 7.28 | 20.03 | 12.82 |
| ViT | **4.78** | **11.87** | **8.59** | 4.59 | **13.40** | 10.58 |
| DINO | 5.16 | 14.53 | 11.23 | **4.02** | 25.35 | **7.66** |
| MAE | 15.15 | 38.72 | 34.26 | 10.08 | 62.17 | 18.85 |

## 4.3 Scaling Up the Proposal Network

We examine the IG performance of tokenizers with a large-scale proposal network. Following VQGAN, we adopt GPT-2 XL as $\mathcal{P}_{AR}$, which comes with 1.4B parameters. In line with Tab. 1, VQ-KD$_{ViT}$ leads with 9.23 FID$_{AR}$, while VQ-KD$_{CLIP}$ achieves the highest IS metric at 150.63. Upon comparing with Tab. 1, tokenizers with stronger IU capabilities exhibit less improvement in the FID$_{AR}$ metric. For instance, VQ-KD$_{MAE}$ improves significantly from 26.85 to 17.11, while VQ-KD$_{CLIP}$ reveals a marginal enhancement from 11.78 to 11.27. This suggests that a small-scale $\mathcal{P}_{AR}$ is sufficient for tokenizers with strong IU capabilities, whereas those with weaker IU abilities benefit from a large-scale $\mathcal{P}_{AR}$.

Table 5: AR modeling with a large-scale proposal network or strong data augmentation.

| Tokenizer $\mathcal{T}$ | GPT-2 XL | | Strong Aug. | |
|---|---|---|---|---|
| | FID$_{AR}$ ↓ | IS$_{AR}$ | FID$_{AR}$ ↓ | IS$_{AR}$ |
| VQGAN | 17.13 | 59.19 | 32.76 | 26.76 |
| FSQ | 25.87 | 49.59 | 52.17 | 18.09 |
| VQ-KD$_{CLIP}$ | 11.27 | **150.63** | 15.24 | **90.41** |
| VQ-KD$_{ViT}$ | **9.23** | 146.00 | **13.32** | 81.44 |
| VQ-KD$_{DINO}$ | 9.50 | 120.26 | 19.39 | 52.74 |
| VQ-KD$_{MAE}$ | 17.11 | 69.03 | 36.63 | 28.49 |
| Cluster$_{CLIP}$ | 14.00 | 110.26 | 17.22 | 83.72 |

## 4.4 Influence of Strong Data Augmentation

We investigate the impact of strong data augmentation on the AR modeling performance of tokenizers. Specifically, we employed a strong random crop, where the crop scale ranges from 0.08 to 1.0, introducing greater variability into the training data. As shown in Tab. 5, all tokenizers exhibit worse FID$_{AR}$ metrics than their counterparts in Tab. 1. Interestingly, tokenizers with stronger IU capabilities demonstrate greater robustness to the strong data augmentation. For instance, VQ-KD$_{ViT}$ experiences a minor increase in FID$_{AR}$ of just 1.92 (from 11.40 to 13.32), whilst VQ-KD$_{MAE}$ records a considerable leap of 9.78 (from 26.85 to 36.63).

## 4.5 Large Teacher Models in VQ-KD

We incorporate OpenCLIP [17] models of varying sizes as teacher models to train the VQ-KD tokenizers. As illustrated in Tab. 6, the FID$_{AR}$ metric sees a reduction from 10.31 to 8.70 when the size of the OpenCLIP model escalates from ViT-L/14 to ViT-G/14. Given that larger Open-CLIP models inherently possess stronger IU capabilities, these findings further corroborate the superiority of image tokenizers with more potent IU capabilities.

Table 6: Effect of different teachers in VQ-KD.

| OpenCLIP | rFID ↓ | PPL ↓ | FID$_{AR}$ ↓ | IS$_{AR}$ |
|---|---|---|---|---|
| ViT-L/14 | 4.03 | 80.56 | 10.31 | 146.21 |
| ViT-H/14 | **3.60** | 97.32 | 9.64 | **161.13** |
| ViT-G/14 | 3.80 | **77.79** | **8.70** | 152.71 |

Table 7: Effect of codebook size and dimension. Experiments are conducted on VQ-KD$_{CLIP}$.

(a) Codebook size.

| Codebook **C** | | rFID ↓ | FID$_{AR}$ ↓ | IS$_{AR}$ |
|---|---|---|---|---|
| Size (log$_2$) | Usage (%) | | | |
| 10 | **100.0** | 6.59 | 11.65 | 114.90 |
| 11 | **100.0** | 5.99 | **10.98** | 119.72 |
| 12 | **100.0** | 5.64 | 11.71 | 123.42 |
| 13 | **100.0** | 4.96 | 11.78 | 128.18 |
| 14 | 93.7 | **4.53** | 11.61 | **131.51** |

(b) Codebook dimension.

| Codebook **C** | | rFID ↓ | FID$_{AR}$ ↓ | IS$_{AR}$ |
|---|---|---|---|---|
| Dim | Usage (%) | | | |
| 16 | **100.0** | 4.94 | 12.19 | 124.71 |
| 32 | **100.0** | 4.96 | 11.78 | **128.18** |
| 64 | 97.4 | **4.64** | 11.00 | 126.60 |
| 128 | 89.6 | 5.03 | **10.50** | 119.78 |
| 256 | 48.7 | 6.80 | 12.08 | 103.07 |

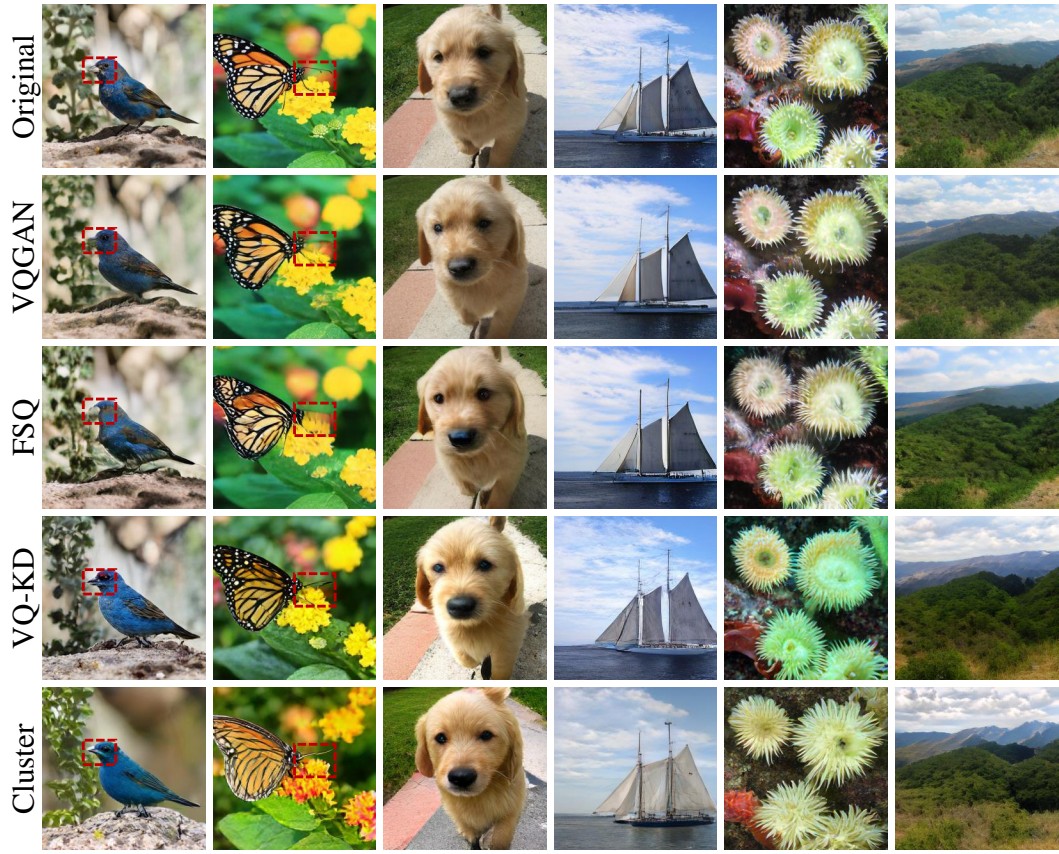

Figure 5: Reconstruction results of different image tokenizers.

## 4.6 Codebook Size and Dimension

The size and dimension of the codebook exert a significant influence on the IG performance of tokenizers [10, 47]. Tab. 7a showcases the performance of VQ-KD$_{CLIP}$ with varying codebook sizes. Large codebooks aid the tokenizers in representing fine-grained semantics, contributing to a consistent decrease in the rFID metric from $6.59$ to $4.53$. The IS metric also shows favor towards larger codebooks, with size $2^{14}$ leading to the highest IS metric of $131.51$. However, choosing the correct code from a large codebook is harder than from a small codebook, hindering $\mathcal{P}_{AR}$ from achieving lower FID scores with larger codebooks.

Tab. 7b demonstrates the influence of codebook dimension. High-dimensional codes carry more information but lead to lower codebook usage. As a result, the rFID metric initially drops from $4.96$ to $4.64$, then increases drastically to $6.80$. Similar to Tab. 7a, the FID and IS metrics favor different codebook dimensions. FID favors $128$-dimensional codebooks, where codebook usage is relatively low. In contrast, IS favors $32$-dimensional codebooks, possibly due to a superior diversity.

## 4.7 Qualitative Analysis

The reconstruction quality of various tokenizers is demonstrated in Fig. 5. Original images are displayed in the first row. Regions where VQGAN and FSQ fail to reconstruct are highlighted with red boxes. In contrast, VQ-KD reconstructions are visually more accurate. Since the IU encoder in Cluster emphasizes encoding semantics over visual details, Cluster fails to preserve all visual details during reconstruction. Nonetheless, the reconstruction results of Cluster still appear more natural than VQGANand FSQ, especially in the highlighted areas. Fig. 6 and Fig. 7 further illustrates the AR and NAR generation results, showcasing the superior visual performance of VQ-KD and Cluster.

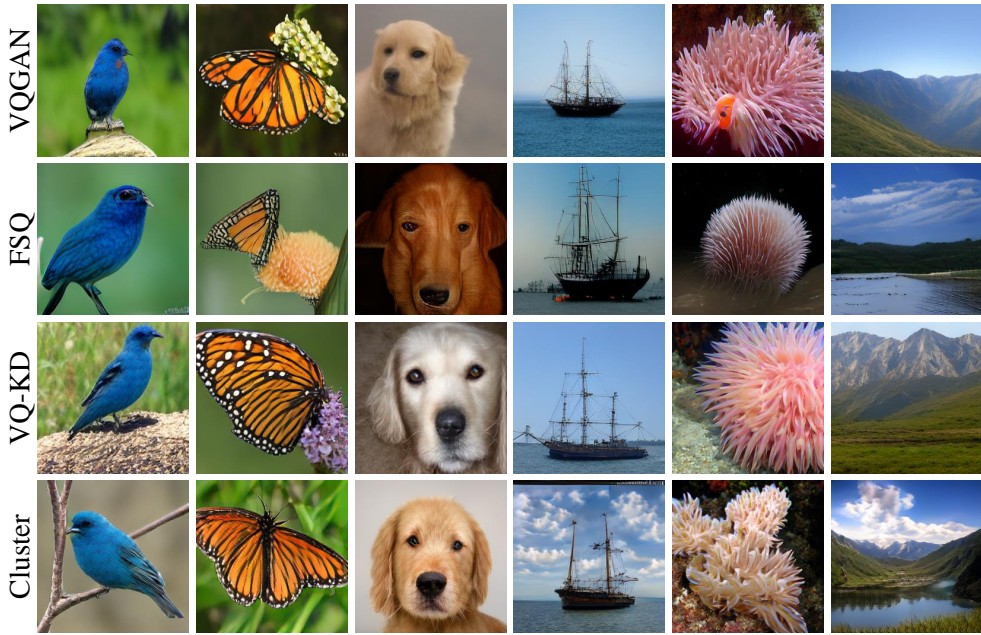

Figure 6: Class-conditional AR generation results of different image tokenizers.

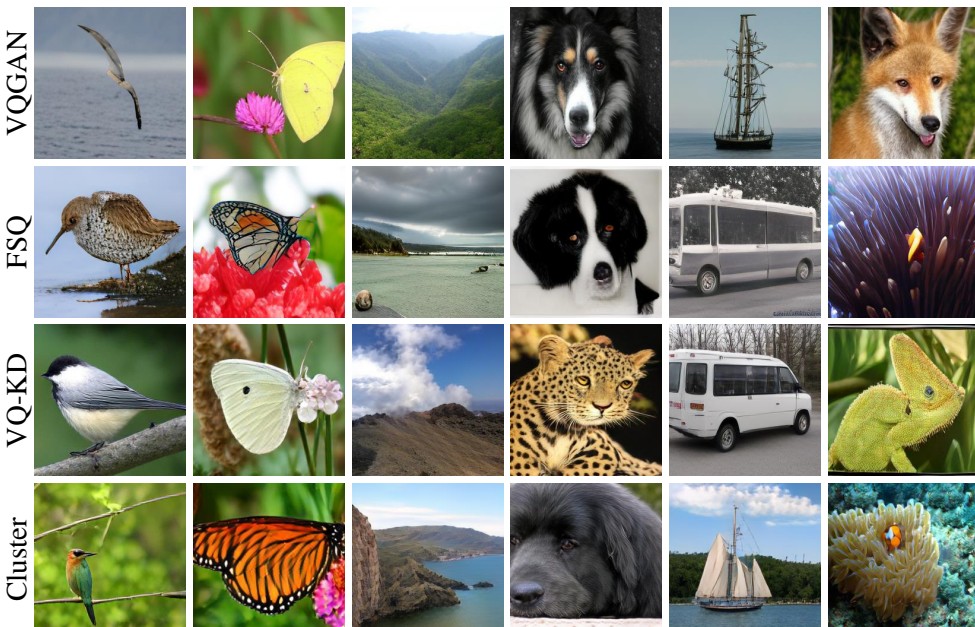

Figure 7: Unconditional NAR generation results of different image tokenizers.

## 5 Conclusion

In this paper, we show that image understanding (IU) models can be useful in image generation (IG). Through comprehensive studies, observe that the VQ-KD tokenizers significantly enhance generation quality over VQGAN, irrelevant of the quantization operation and codebook utilization. Within the VQ-KD tokenizers, stronger IU capabilities tends to deliver superior IG performance. Further verification shows that the superiority of VQ-KD holds across proposal networks, datasets, and tasks. Lastly, we analyze VQ-KD from multiple angles, including clustering pretrained models as tokenizers, scaling up the proposal network, influence of strong data augmentation, large teacher models in VQ-KD, and codebook size and dimensions.

## Acknowledgement

This research is supported in part by National Key R&D Program of China (2022ZD0115502), National Natural Science Foundation of China (NO. 62122010, U23B2010), Zhejiang Provincial Natural Science Foundation of China under Grant No. LDT23F02022F02, Key Research and Development Program of Zhejiang Province under Grant 2022C01082, "Pioneer" and "Leading Goose" R&D Program of Zhejiang (No. 2024C01161).

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

# A  Datasets

The experiments are conducted on two image datasets: ImageNet-1k (IN-1k) [7] and MS-COCO [23]. The IN-1k dataset contains approximately $1.28$ million training images and $50,000$ validation images across $1,000$ diverse categories. The MS-COCO dataset comprises $82,783$ images for training and $40,504$ for validation. Each image is annotated with several captions.

For a given image, we first resize its shorter side to $s$ pixels, where $s$ symbolizes the input size. Subsequently, a central crop is performed to derive an image fragment sized $s \times s$ pixels. Our default augmentation strategy incorporates a random crop (between 0.8 and 1.0) partnered with random horizontal flipping.

# B  Implementation Details

The image tokenizers under consideration generate token sequences of length 256 upon a $256 \times 256$ input image. All tokenizers remain frozen throughout the training of the decoder and the proposal networks. Experiments are performed using 8 A100 80GB GPUs. The approximate training times for the VQ-KD tokenizer is around 30 hours, the decoder requires roughly 68 hours, while the AR proposal network and NAR proposal network necessitate about 29 hours and 72 hours, respectively. In total, a single experiment takes approximately 200 hours training.

The CNN-based VQGAN tokenizers, with 27.9M parameters, are trained following the identical procedure employed for decoders. The codebook dimension of VQGAN is 256. FSQ levels $\mathcal{L}$ are set to $(8, 8, 5, 5, 5)$, equivalent to codebook size $8,000$.

As per BEiT v2 [27], an AdamW optimizer is utilized to train the VQ-KD tokenizers. The learning rate warms up linearly to $10^{-4}$ for $25,000$ steps, subsequently decaying to $10^{-5}$ under a cosine schedule. Unless specifically stated, VQ-KD tokenizer is trained with an input size of $224 \times 224$ and codebook dimension of 32.

VQGAN [10] is adopted for training both the decoder and the AR proposal networks. Both $\mathcal{D}$ and $\mathcal{P}_{AR}$ training span $260,000$ steps with a collective batch size of 96 for IN-1k and 24 for MS-COCO. The decoder is a CNN-based VQGAN decoder, consisting of 40.5M parameters. The decoders utilize the Adam [18] optimizer with learning rates set at $5.4 \times 10^{-5}$, $\beta_1 = 0.5$, and $\beta_2 = 0.9$. Their discriminators are also trained via Adam optimizer, employing learning rates of $4.32 \times 10^{-4}$, while keeping the $\beta$ constants identical. Subsequent training of AR proposal networks relies on the AdamW [24] optimizer with $\beta_1 = 0.9$, $\beta_2 = 0.98$, and a $0.2$ weight decay. An initial learning rate of $10^{-4}$ is set, after which it decays to 0 on a cosine schedule. The AR proposal network is a GPT-2 Medium [31], with 335M parameters.

We follow MAGE [22] for training NAR proposal networks. $\mathcal{P}_{NAR}$ is trained for 300 epochs with a collective batch size of 512 on ImageNet-1k. NAR proposal networks are trained with the AdamW optimizer with $\beta_1 = 0.9$, $\beta_2 = 0.95$, and a $0.05$ weight decay. The learning rate warms up linearly to $3 \times 10^{-4}$ throughout 10 epochs before decaying to 0 following a cosine schedule. The encoder of $\mathcal{P}_{NAR}$ is a ViT-B/16, with 86M parameters.

# C  Evaluation

Codebook usage is defined as the proportion of codes from the codebook that have been used at least once when encoding the dataset. A low value for codebook usage might be an indication of the 'codebook collapse' issue.

IS provides a measure of both the fidelity and diversity of $\tilde{\mathbf{I}}$. However, IS significantly relies on the classification capabilities of a pretrained Inception-v3 model [38]. Complex images are likely to be misinterpreted as lacking fidelity by IS. Therefore, we limit the use of IS to IN-1k experiments only.

To circumvent the limitations of IS, FID computes the statistical distance in the Inception-v3 feature space between the real images $\mathbf{I}$ and the generated images $\tilde{\mathbf{I}}$. A lower FID score indicates that $\tilde{\mathbf{I}}$ is statistically similar to $\mathbf{I}$.

rFID is defined as the FID between $\mathbf{I}$ and their reconstructed counterparts $\hat{\mathbf{I}}$. Obtaining a low rFID score requires that the image tokenizer encode sufficient visual details within the codes $\mathbf{C}(\mathbf{z})$ to enable accurate reconstruction by the decoder.

The PPL score is defined as:

$$\text{PPL} = \exp\left(-\frac{1}{L}\sum_{i=1}^{L}\log p(z_i|z_{1:i-1})\right), \tag{5}$$

where $\mathbf{z}$ denotes a sequence of codes offered by the tokenizer, $L$ represents the length of $\mathbf{z}$, and $p(\mathbf{z})$ embodies the distribution modeled by the AR proposal network $\mathcal{P}_{\text{AR}}$.

Both reconstruction and AR modeling serve as two pivotal capabilities in an image generator. We anticipate that these metrics will lead to a more thorough insight into the generative capacities of image tokenizers.

## D Limitations

The VQ-KD tokenizers are designed to mimic the IU encoders, yielding superior quantitative results compared to traditional tokenizers like the VQGAN and FSQ. Nonetheless, qualitative analysis suggest that the VQ-KD may modify visual details during the pixel reformation process, thereby posing challenges for tasks such as image editing.

## E Broader Impacts

This paper explores the question *how might image understanding (IU) models aid image generation (IG) tasks*. We envision that our findings will motivate research on image tokenizers and prompt the community to reconsider the correlation between IU and IG.

