# OpenReview forum: "Image Understanding Makes for A Good Tokenizer for Image Generation"
_NeurIPS.cc/2024/Conference — NeurIPS 2024 poster_

### Official Review · Reviewer_WgNi · 2024-07-11

**Soundness:** 3
**Presentation:** 2
**Contribution:** 2
**Rating:** 6
**Confidence:** 3

**Summary:**

This paper shows that image understanding models can be helpful in image generation and that a stronger IU model can result in better IG performance. To demonstrate it, this paper sets multiple experiments from many perspectives (e.g. different datasets and codebook sizes) and gives out reasonable analysis.

**Strengths:**

1.This paper has done experiments across various perspectives, making a solid demonstration.

2.The results of different VQ-KD models outperforming VQGAN and FSQ support this paper's standpoint.

3.The visualization of codebooks is very concise, clear and well drawn.

4.The idea of training a clustering based tokenizer from an IU model is interesting, which can also illustrate the standpoint.

**Weaknesses:**

1.The visualization of reconstruction of VQ-KD and Cluster seems worse than that of VQGAN and FSQ in both paper and appendix from my point of view. Maybe more visual results should be included.

2.This paper seems to claim that IU model can help IG, but this paper's experiments and analysis are limited to token based tasks. I think this paper should include more IG methods like diffusion, VAE with some little experiments or analysis if possible. Or this paper should emphasize the range is limited to token based tasks.

3.Various experiments have been done across different settings and perspectives, but I think this paper lacks of methods.Besides VQGAN, FSQ and VA-KD, more related methods should be included.

4.The structure of this paper is a little confusing, with experiments, analysis and descriptions mixed together.

**Questions:**

1.Why doesn't a larger teacher model always lead to better results? Following the theory of this paper, ViT-G/14 should perform best.

2.The experiment about different codebook sizes and dimensions seems to reveal that when the IU model gets stronger, the IG may not always be better. Can I think that to make better IG results, we don't have to use an IU model with too strong capability? I need detailed analysis.

**Limitations:**

From this paper's limitations, although VQ-KD performs better due to its stronger capability in image understanding, it is not suitable for all scenarios.

---

> ### Author Rebuttal · Authors · 2024-08-07
>
> Thank you for your detailed review and valuable comments. Please note our top-level comment. Below we address specific questions.
>
> # Qualitative results
>
> Please see our top-level comment. We present more qualitative results in the attached pdf file. Note that image reconstruction capabilities and image generation capabilities may not be aligned. Our primary focus is on image generation.
>
> # Generalizability
>
> The primary focus of this paper lies in the AutoRegressive (AR) IG framework (as mentioned by line 27 in the main text). We are revising our manuscript to further clarify the scope of our work.
>
> We also conduct preliminary experiments on Diffusion Models and observe that IU capabilities may enhance their performance. We initially follow LDM [9] to train a VQGAN and a VQ-KD CLIP model. Both models adopted 8-dimensional codebooks of size 16384. Subsequently, we trained diffusion models for 20 epochs on IN-1k for conditional image generation. We summarize the performance of VQGAN, VQ-KD CLIP, and their corresponding diffusion models below:
>
> | Tokenizer | Codebook Usage (%) | rFID | Diffusion Model FID |
> |---|---|---|---|
> | VQGAN | 2.0 | 5.70 | 31.67 |
> | VQ-KD | 100.0 | 5.47 | 31.31 |
>
> We find that the FID score of the VQ-KD model (at 31.31) is marginally lower than the FID score of the VQGAN model (at 31.67). Yet, it is important to note that the current VQ-KD and Cluster methods are tailored for token-based generators, because token-based frameworks are more apt for exploring the interplay between Image Understanding (IU) and Image Generation (IG). Consequently, these methods may not be directly applicable to continuous VAE models. We remain on the lookout for alternative frameworks that could effectively incorporate IU to assist continuous VAE models.
>
> # Comparison Experiments
>
> Currently, image tokenization methods can be roughly separated into two groups: vector-quantization-based and scalar-quantization-based. VQGAN adopts vector quantization, while FSQ is one of the latest image tokenizers that adopt scalar-quantization. Other works like SQ-VAE and CVQ-VAE introduce various improvements to VQGAN, but share the same architecture as VQGAN, so their impact on our final conclusion is little.
>
> For comparison with related works, we adopt a stronger recipe to train the AR proposal network, with larger batch size (32 per GPU), more training epochs, and classifier-free-guidance (CFG). As shown in the following table, VQ-KD CLIP outperforms prior image tokenizers:
>
> | Image Tokenizer | FID |
> |---|---|
> | RQVAE [3] | 10.38 |
> | ViT-VQGAN [4] | 4.17 |
> | MoVQ [5] | 8.78 |
> | MaskGIT [6] | 4.51 |
> | FSQ [7] | 4.53 |
> | VQ-KD CLIP | 4.10 |
>
> # Presentation
>
> Our work studies the relationship between IU and IG. We adopt this organization to clearly express our findings. Any better suggestions are welcome.
>
> # Larger Teacher Models
>
> In Tab. 6 of the main text, as the OpenCLIP teacher gets larger, the FID AR metric consistently decreases. The FID AR metric evaluates the similarity between generated images and reference images (usually the IN1k val split), so we use FID AR as the primary metric. Other metrics including rFID, PPL, and IS AR, evaluate the reconstruction quality, quality of token sequence modeling, and diversity of the generated images. These metrics may be subject to noise disturbance, so a larger teacher does not guarantee better metrics. Nonetheless, larger teachers do tend to yield better metrics.
>
> # Codebook Sizes and Dimensions
>
> Larger codebook sizes or dimensions do not necessarily lead to stronger IU models. For instance, in Tab. 7 (b) of the main text, increasing the codebook dimension leads to a significant drop in codebook usage, which limits the IU capabilities of VQ-KD.
>
> To further investigate the relationship between IU and IG capabilities, we conduct linear probing experiments on IN1k, where tokenizers with higher Top-5 Acc. metrics tend to achieve better FID AR metrics:
>
> |  | PPL AR $\downarrow$ | FID AR $\downarrow$ | Top-5 Acc. |
> |---|---|---|---|
> | VQ-KD CLIP | 53.73 | 11.78 | 75.11 |
> | VQ-KD ViT | 89.30 | 11.40 | 64.88 |
> | VQ-KD DINO | 74.07 | 13.15 | 54.17 |
> | VQ-KD MAE | 280.06 | 26.85 | 41.39 |
>
> The above results suggest that tokenizers with stronger IU capabilities tend to behave better in IG tasks.

---

> ### Comment · Reviewer_WgNi · 2024-08-12
>
> I have carefully read your rebuttals and the rebuttals have solved my questions. The new visualizations can reflect that IU model helps IG tasks well. The experiments on diffusion models can also demonstrate this paper's conclusion(though diffusion is not actually what this paper discusses). The comparisons and linear probing experiments about IU capability are clear. And it is good for you to do such detailed experiments during this time. I think this paper is worth accepted.

---

> > ### Author Response · Authors · 2024-08-13
> >
> > Thank you so much for your detailed response and raising your score.

---

### Official Review · Reviewer_3Kzg · 2024-07-12

**Soundness:** 3
**Presentation:** 3
**Contribution:** 3
**Rating:** 5
**Confidence:** 4

**Summary:**

This work focuses on the connection between image understanding (IU) and image generation (IG). The authors introduce a token-based IG framework and a novel feature reconstruction objective for tokenizer training. They introduce an extra feature reconstruction loss to distill semantic knowledge from pretrained IU model to tokenizer. They demonstrate superior IG performance with such tokenizers having strong IU capabilities, as evidenced by various metrics, datasets, tasks, and networks.

**Strengths:**

- The paper is well-written, systematically organized, and straightforward to follow.
- The idea to merge the image understanding and generation model is sensible and is shown to be effective.
- Plenty of quantitative experiments are carried out to validate the superiority of the proposed method.

**Weaknesses:**

1. **Excessive and unjustified claims.**
    - An important contribution stated by this work is that it is the first to combine IU with IG, which might not hold true. I am not very familiar with related works in this specific direction, but I can mention one from ICLR 2024 [a]. They not only use IU as a tokenizer but also reduce the token number in a dynamic way.
1. **Observations in Sec. 3.4 needs further justification** For example,
    - More experiments are needed to verify that *The superiority of VQ-KD is irrelevant to the quantization operation and codebook usage.*
    - The observation that *Tokenizers with stronger semantic understanding tend to deliver superior IG performance* are not fully supported  by the experimental results.

1. **More visualization results are necessary.**
    - The paper is almost completely backed up by quantitative results. This may not be favorable for an image generation method. Many more examples are needed to enhance the persuasiveness of the experiment. For example, results on MSCOCO (Tab. 3), qualitative comparison with different IU backbone, etc.

1. **Lack of qualitative analysis for the learned codebook.**
    - There is no visualization of what is encoded in the VQ-KD and why it is superior to the original VQGAN. The authors could refer to [a] for some examples.

1. **Others**
    - Please use `Tab. X` and `Fig. X` in main text.
    - What is $D$ is used in Eq (4)?

[a] Unified Language-Vision Pretraining in LLM with Dynamic Discrete Visual Tokenization. Jin et al. ICLR2024.

**Questions:**

In summary, this is a work featuring good motivation and quantitative results. However, I have the following concerns:
- As I am not familiar  with specific related works, I am unable to offer a competent evaluation of the quantitative results and assess the significance of these outcomes. I would prefer to see the comments from more expert reviewers before arriving at a final decision.
- The comparison with related works appears insufficient. There is almost no comparison made with state-of-the-art image generation models.
- Why are there so few qualitative results? For an image generation paper, visual examples are of considerable importance.
- Visual analysis of learned codebook is also missing.

**Limitations:**

Although the authors discuss the limitations in a few sentences, I believe more should be expounded.

---

> ### Author Rebuttal · Authors · 2024-08-07
>
> Thank you for your detailed review and thoughtful comments. Please note our top-level comment. Below we address specific questions.
>
> # Related works
>
> Our main conclusion is that IU models can aid IG tasks, which have not been explored before. While LaViT adopts a pretrained ViT encoder in the tokenizer, they did not perform in-depth analysis regarding the relationship between IU and IG. We will modify our claim and clarify that we are the first to demonstrate that IU models can substantially enhance IG through VQ-KD.
>
> # Main observations
>
> We perform an ablation study on the quantization operation of VQ-KD and show the results in our top-level comment. The results suggest that VQ-KD can optimize the quantizer in the same way as VQGAN, while achieving better IG performance.
>
> As for codebook usage, Tab. 7 (b) in the main text shows that when the codebook dimension of VQ-KD increases to 256, which is the same as VQGAN, the codebook usage of VQ-KD drops to 48.7%. However, VQ-KD still achieves 12.08 FID AR, which is significantly better than the 24.11 FID AR metric of VQGAN.
>
> To support the conclusion that Tokenizers with stronger semantic understanding tend to deliver superior IG performance, we further conduct linear probing experiments on IN1k:
>
> |  | PPL AR $\downarrow$ | FID AR $\downarrow$ | Top-5 Acc. |
> |---|---|---|---|
> | VQ-KD CLIP | 53.73 | 11.78 | 75.11 |
> | VQ-KD ViT | 89.30 | 11.40 | 64.88 |
> | VQ-KD DINO | 74.07 | 13.15 | 54.17 |
> | VQ-KD MAE | 280.06 | 26.85 | 41.39 |
>
> As shown in the above table, VQ-KD CLIP and VQ-KD ViT achieve the highest Top-5 Acc. scores, while VQ-KD MAE achieves the worst Top-5 Acc metric. This trend in the Top-5 Acc. metric is roughly the same as the trend in the FID AR metric, suggesting that tokenizers with stronger IU capabilities tend to behave better in IG tasks.
>
> # Qualitative results
>
> Please see our top-level comment. We present more visualizations and quantitative analysis for the VQ-KD codebook in the attached PDF file.
>
> # Presentation
>
> Thanks for the suggestions. We are revising our manuscript and will use Tab./Fig. for cross-reference. In Eq. (4), the symbol $\mathcal{D}$ represents the decoder of VQ-KD, which maps the code vectors $\mathbf{C}(z)$ to the feature space of teacher $\mathcal{T}'$. We apologize for the confusion and will provide additional information in the main text.
>
> # Comparison Experiments
>
> We adopt a stronger recipe to train the AR proposal network, with larger batch size (32 per GPU), more training epochs, and classifier-free-guidance (CFG). As shown in the following table, VQ-KD CLIP outperforms prior AR and NAR methods, and is comparable to some Diffusion-based methods.
>
> |  | Architecture | FID |
> |---|---|---|
> | RQVAE [3] | AR | 10.38 |
> | ViT-VQGAN [4] | AR | 4.17 |
> | MoVQ [5] | NAR | 8.78 |
> | MaskGIT [6] | NAR | 4.51 |
> | FSQ [7] | NAR | 4.53 |
> | ADM [8] | Diffusion | 4.59 |
> | LDM-4-G [9] | Diffusion | 3.60 |
> | CVQ-VAE [10] | Diffusion | 6.87 |
> | DiT-XL/2-G [11] | Diffusion | 2.27 |
> | VQ-KD CLIP | AR | 4.10 |

---

> ### Author Response · Authors · 2024-08-13
>
> Dear Reviewer,
>
> Thank you for the suggestions that help us improve the paper. As the deadline for discussion is approaching, please let us know if you have any additional questions. We genuinely hope you can consider raising the score if we have satisfactorily addressed your concerns.
>
> Thanks again,
>
> The Authors

---

### Official Review · Reviewer_aFBf · 2024-07-12

**Soundness:** 3
**Presentation:** 4
**Contribution:** 3
**Rating:** 5
**Confidence:** 3

**Summary:**

This paper introduces a novel framework that leverages the rich semantic capabilities of Image Understanding models for Image Generation tasks. By employing a token-based generation framework and a feature reconstruction objective, the paper trains tokenizers capable of mapping images into token sequences. Compared to traditional pixel reconstruction methods, this approach demonstrates good performance across various metrics.

**Strengths:**

The paper's strength lies in its innovative fusion of image understanding with image generation frameworks, offering a fresh perspective that transcends traditional pixel-based approaches. The originality is evident in the novel application of feature reconstruction for tokenizer training, drawing knowledge from pre-trained image understanding models, which is a creative synthesis of existing concepts. The significance of this work is underscored by its potential to redefine tokenizer research and enhance generation performance across various metrics, as demonstrated through rigorous empirical validation on the ImageNet-1k dataset. The clarity of the paper is reflected in its well-structured presentation and articulate explanation of complex concepts, making the methodology and results easily comprehensible. Overall, the quality of the research, its original problem formulation, and the clarity of the findings contribute to the paper's impact and broad applicability.

**Weaknesses:**

1. The paper lacks theoretical support for the VQ-KD tokenizer.
2. The experiments are limited to the ImageNet and COCO dataset, which may not be sufficient to demonstrate the effectiveness of the method. Consider adding higher resolution and higher quality datasets, like the LAION-Aesthetics
3. There is a lack of comparison and discussion regarding computational efficiency and computational load.

**Questions:**

Please see weaknesses

**Limitations:**

1. Experiments dataset is limited
2. Generalizability was not enough explored and discussed

---

> ### Author Rebuttal · Authors · 2024-08-07
>
> Thank you for your constructive comments. Please note our top-level comment. Below we address specific questions.
>
> # Theoretical Support
>
> While it is hard to theoretically prove the superiority of the VQ-KD tokenizer, we hypothesize that superiority is because each token generated by VQ-KD contains rich semantics, rather than simply appearance information. The hypothesis is supported by the finding that the PPL AR metric of most VQ-KD tokenizers are lower than VQGAN and FSQ in Tab. 1 of the main text. Lower PPL AR metric suggests that the proposal network can easily model the token sequence generated by VQ-KD. Moreover, Fig. 4 in the main text demonstrates that images belonging to the same category are encoded with similar tokens. Please also refer to Fig. 2 in the attached PDF file for a qualitative analysis of the semantics encoded within each VQ-KD token.
>
> # Generalizability
>
> Please see our top-level comment. We perform experiments on challenging datasets including LAION-Aesthetics. Visualizations are shown in the attached PDF file. The results demonstrate that our conclusion generalizes well to various datasets.
>
> # Computation cost
>
> We present the computation cost for each training stage of VQGAN and VQ-KD in the following tables. All experiments are conducted on IN-1k using 8 A100-80G GPUs. In sum, VQGAN takes 81 hours to train and VQ-KD CLIP takes 89 hours to train. The training cost for VQ-KD CLIP is only 10% higher than VQGAN.
>
> | VQGAN | #epochs | CUDA Memory (GB) | Training Time (hours) |
> |---|---|---|---|
> | Stage 1: Tokenizer | 20 | 25.6 | 42 |
> | Stage 2: AR Proposal Network | 20 | 17.9 | 39 |
>
> | VQ-KD | #epochs | CUDA Memory (GB) | Training Time (hours) |
> |---|---|---|---|
> | Stage 1: Tokenizer | 100 | 10.7 | 14 |
> | Stage 2: Pixel Decoder | 20 | 18.2 | 36 |
> | Stage 3: AR Proposal Network | 20 | 17.8 | 39 |

---

> > ### Comment · Reviewer_aFBf · 2024-08-12
> >
> > I really appreciate the author's additional experiments on additional datasets, which better prove the advantages of the method. The author addressed most of my concerns, but I still have some concerns about the explanation of VQ-KD tokenizer.
> > I think it is better to keep the original rating.

---

> > > ### Author Response · Authors · 2024-08-13
> > >
> > > Dear Reviewer,
> > >
> > > Thank you for your feedback on our rebuttal. We appreciate your thorough review and would like to ensure we address any remaining concerns you may have. Could you kindly specify which kind of explanation would be most helpful in better supporting the effectiveness of the VQ-KD tokenizer?
> > >
> > > Thank you once again for your valuable insights.
> > >
> > > Best regards,
> > > The Authors

---

### Official Review · Reviewer_cC74 · 2024-07-17

**Soundness:** 4
**Presentation:** 3
**Contribution:** 3
**Rating:** 6
**Confidence:** 3

**Summary:**

This paper explores using image understanding (IU) models to aid image generation (IG) performance. To verify the hypothesis, the authors focus on the different tokenizers and introduce feature reconstruction (VQ-KD) as a training objective for image tokenizers, distilling knowledge from pre-trained IU encoders. This paper compares VQ-KD tokenizers with conventional methods like VQGAN and FSQ across various metrics, datasets, tasks, and proposal networks. The results show that tokenizers with strong IU capabilities, particularly VQ-KD, outperform traditional methods.

**Strengths:**

+ The paper is the first to demonstrate that image understanding models can substantially enhance image generation.
+ To verify the hypothesis, the analyses of different tokenizers and training objectives are reasonable.
+ The authors conduct extensive experiments across different metrics, datasets, tasks, and network architectures to validate their findings.
+ The paper gives detailed visualizations and analyses that help in understanding why the proposed model outperforms existing methods.
+ Experiments show the usage of VQ-KD tokenizers outperforms conventional methods, achieving state-of-the-art results on several benchmarks.

**Weaknesses:**

- While the paper examines four types of IU encoders, it could benefit from investigating a broader range of IU models to strengthen its conclusions.
- The paper could include more detailed ablation studies to isolate the impact of different components in the VQ-KD approach.
- As in the limitation, maybe it is better to introduce the fidelity-related metrics to indicate that this work has a better generation ability, but may result in lower fidelity.
- Some visual comparisons are not obvious. It is better to give a close-up to see the details.

**Questions:**

1. How does the proposed method perform on more diverse and challenging datasets beyond ImageNet-1k and MS-COCO, such as medical imaging or satellite imagery? This may show the generalization ability of the proposed model.
2. The description of VQGAN 'pixel reconstruction' seems not suitable, as VQGAN adopts the perceptual loss which may also introduce the semantic constraint.

**Limitations:**

Yes.

---

> ### Author Rebuttal · Authors · 2024-08-07
>
> Thank you for your thoughtful feedback. Please note our top-level comment. Below we address specific questions.
>
> # Generalizability
>
> Please see our top-level comment. We examine VQ-KD ConvNext and observe good image generation abilities. More types of IU encoders will be added to our revised manuscript. We also conduct VQGAN and VQ-KD CLIP experiments on medical imaging (SA-Med2D-20M) and satellite imagery (SATIN) datasets, where VQ-KD CLIP consistently outperforms VQGAN. Visualizations are shown in the attached PDF file. These results confirm the generalization ability of our conclusions across different models and datasets.
>
> # Ablation Study
>
> Please refer to our top-level comment, where we replace the K-Means module in VQ-KD with the quantization loss used in VQGAN. A slight performance drop is observed, but the overall performance of VQ-KD still surpasses VQGAN by a large margin.
>
> # Fidelity-Related Metrics
>
> Some results in Tab. 7 of the main text shows that the generation ability of VQ-KD can surpass VQGAN even if its rFID metric is relatively lower. To further support this observation, we conduct experiments with a VQ-KD CLIP model that is not well-trained. As shown in the following table, while the rFID metric of this VQ-KD CLIP model is worse than VQGAN, its PPL and FID metrics still outperform VQGAN by a large margin.
>
> |  | rFID $\downarrow$ | PPL $\downarrow$ | FID $\downarrow$ |
> |:-:|---|---|---|
> | VQGAN | 5.09 | 116.75 | 24.11 |
> | VQ-KD CLIP (codebook size 1024) | 6.59 | 21.28 | 11.65 |
> | VQ-KD CLIP (codebook dim 256) | 6.80 | 16.44 | 12.08 |
> | VQ-KD CLIP (not well-trained) | 5.26 | 62.77 | 11.79 |
>
> # Qualitative results
>
> Please see our top-level comment. We present more qualitative results in the attached pdf file.
>
> # VQGAN description
>
> Thanks for pointing out this ambiguity. We will clarify this in our revised manuscript to avoid confusion. VQGAN adopts perceptual loss to enhance the perceptual quality of reconstructed images. Given a real image $\mathbf{I}$ and a reconstructed image $\hat{\mathbf{I}}$, the perceptual loss encodes both images and outputs the distance between the image features. While semantic constraint is introduced, the primary goal of perceptual loss is to achieve better pixel reconstruction.

---

> ### Author Response · Authors · 2024-08-13
>
> Dear Reviewer,
>
> Thank you for the suggestions that help us improve the paper. As the deadline for discussion is approaching, please let us know if you have any additional questions. We genuinely hope you can consider raising the score if we have satisfactorily addressed your concerns.
>
> Thanks again,
>
> The Authors

---

> > ### Comment · Reviewer_cC74 · 2024-08-14
> >
> > Hi Authors,
> >
> > I do not have other questions. Thanks for addressing my concerns.

---

### Author Rebuttal · Authors · 2024-08-07

Dear reviewers,

We would like to thank you all for providing constructive feedback that helps us improve the paper. We are encouraged by the reviews:

- "The paper is the first to demonstrate that image understanding models can substantially enhance image generation." (Reviewer cC74)
- "The clarity of the paper is reflected in its well-structured presentation and articulate explanation of complex concepts." (Reviewer aFBf)
- "Plenty of quantitative experiments are carried out to validate the superiority of the proposed method." (Reviewer 3Kzg)
- "The visualization of codebooks is very concise, clear and well drawn." (Reviewer WgNi)

We've devoted considerable effort to enhancing our manuscript, and addressing the valuable feedback you've provided. Here, we summarize the key revisions including qualitative results, generalizability, and ablation study. For a more detailed discussion, we encourage you to review our responses to individual reviewer comments.

# Qualitative results

In the attached PDF file, we present more visualizations to demonstrate the superior image generation (IG) capabilities of VQ-KD. The comparison between reconstructed images of VQGAN and VQ-KD CLIP is shown in Fig. 1, with the key regions highlighted. We present qualitative analysis on the codebooks of VQGAN and VQ-KD in Fig. 2. Fig. 3 contains the images generated by VQ-KD CLIP. Lastly, Fig. 4 to Fig. 6 illustrates the reconstructed and generated images on LAION-Aesthetics, SA-Med2D-20M [1], and SATIN [2] datasets.

# Generalizability

## More IU Encoders

In the manuscript, we examined four types of IU encoders (CLIP, DINO, ViT, and MAE), representing fully-supervised, text-supervised, contrastive, and MIM models. To further strengthen our conclusions, we introduce ConvNext as a representative of convolutional IU encoders. Specifically, we use a ConvNext Base model pretrained on IN1k as the teacher to train VQ-KD. The results are shown in the following table, where VQ-KD ConvNext consistently outperforms VQGAN.

|  | rFID | PPL AR | FID AR | IS AR |
|---|---|---|---|---|
| VQGAN | 5.09 | 116.75 | 24.11 | 39.52 |
| VQ-KD ConvNext | 3.57 | 22.20 | 9.68 | 208.10 |

## More Datasets

From the dataset perspective, in addition to IN-1k and MS-COCO, we also perform experiments on three challenging datasets: LAION-Aesthetics, SA-Med2D-20M [1], and SATIN [2]. LAION-Aesthetics is a subset of LAION 5B with high visual quality. SA-Med2D-20M is a large benchmark dataset in the field of medical imaging. SATIN is a metadataset containing 27 constituent satellite and aerial image datasets spanning 6 distinct tasks. Visualizations are shown in the attached pdf file. The tables below present the rFID and FID AR metrics for quantitative comparison. The reference images for FID evaluation are randomly sampled from each dataset.

| LAION-Aesthetics | rFID | FID AR |
|---|---|---|
| VQGAN | 5.98 | 21.19 |
| VQ-KD CLIP | 5.40 | 10.31 |

| SA-Med2D | rFID | FID AR |
|---|---|---|
| VQGAN | 10.64 | 20.46 |
| VQ-KD CLIP | 9.90 | 18.38 |

| SATIN | rFID | FID AR |
|---|---|---|
| VQGAN | 9.75 | 60.89 |
| VQ-KD CLIP | 9.21 | 55.99 |

# Ablation Study

In the main text, we conducted ablation studies of VQ-KD and find that VQ-KD achieves good IG performance with a wide range of teacher models, codebook sizes, and codebook dimensions. In the table below, we further ablate the quantization operation of VQ-KD. As Sec. 3.2 in the main text mentions, VQGAN introduces a quantization loss to optimize the codebook $\mathbf{C}$. In contrast, VQ-KD adopts K-Means to update $\mathbf{C}$. We replace the K-Means module in VQ-KD with quantization loss and observe a slight performance drop. However,  the performance of VQ-KD CLIP w/o K-Means still outperform VQGAN by a large margin, affirming our findings that IU models can be helpful to the IG task.

|  | Codebook Usage | Codebook PPL | rFID | PPL AR | FID AR | IS AR |
|---|---|---|---|---|---|---|
| VQGAN | 4.9 | 5.96 | 5.09 | 116.75 | 24.11 | 39.52 |
| VQ-KD CLIP | 100.0 | 8.93 | 4.96 | 53.73 | 11.78 | 128.18 |
| VQ-KD CLIP w/o kmeans | 100.0 | 8.73 | 5.65 | 43.99 | 12.07 | 72.43 |

---

[1] Sa-med2d-20m dataset: Segment anything in 2d medical imaging with 20 million masks. arXiv preprint arXiv:2311.11969.

[2] Satin: A multi-task metadataset for classifying satellite imagery using vision-language models. arXiv preprint arXiv:2304.11619.

[3] Autoregressive Image Generation using Residual Quantization, CVPR, 2022.

[4] Vector-Quantized Image Modeling With Improved VQGAN, ICLR, 2022.

[5] MoVQ: Modulating Quantized Vectors for High-Fidelity Image Generation, NeurIPS, 2022.

[6] MaskGIT: Masked Generative Image Transformer, CVPR, 2022.

[7] Finite Scalar Quantization: VQ-VAE Made Simple, ICLR, 2024.

[8] Diffusion models beat GANs on image synthesis. NeurIPS. 2021.

[9] High-resolution image synthesis with latent diffusion models. CVPR. 2022.

[10] Online Clustered Codebook, CVPR, 2023.

[11] Scalable diffusion models with transformers. CVPR. 2023.

---

### Comment · Area_Chair_ok7S · 2024-08-12

Dear Reviewers,

This is a reminder that the Reviewer-Author discussion period is nearing its end. We encourage you to participate in the discussion and provide your valuable feedback to the authors. Your comments are greatly appreciated and will contribute to the quality of the submission.

AC

---

### Decision · Program_Chairs · 2024-09-25

**Decision:**

Accept (poster)

**Comment:**

This paper received positive scores of (5, 5, 6, 6) from the reviewers. Reviewers unanimously found the application of image understanding models to enhance image generation intriguing. While concerns were raised regarding the generalizability and qualitative results, the authors successfully addressed these concerns in their rebuttal. The AC, having carefully considered the paper, reviews, and rebuttal, concurs with the reviewers' recommendation for acceptance. The authors are encouraged to incorporate the reviewers' feedback into their final submission.